# Bone Regeneration: A Novel Osteoinductive Function of Spongostan by the Interplay between Its Nano- and Microtopography

**DOI:** 10.3390/cells9030654

**Published:** 2020-03-07

**Authors:** Thomas Vordemvenne, Dirk Wähnert, Julian Koettnitz, Madlen Merten, Nadine Fokin, Andreas Becker, Björn Büker, Asaria Vogel, Daniel Kronenberg, Richard Stange, Günther Wittenberg, Johannes FW Greiner, Andreas Hütten, Christian Kaltschmidt, Barbara Kaltschmidt

**Affiliations:** 1Protestant Hospital of Bethel Foundation, Department of Trauma and Orthopedic Surgery, Burgsteig 13, 33617 Bielefeld, Germany; thomas.vordemvenne@evkb.de (T.V.); dirk.waehnert@evkb.de (D.W.); julian.koettnitz@evkb.de (J.K.); 2Molecular Neurobiology, Bielefeld University, Universitätsstrasse 25, 33615 Bielefeld, Germany; madlen.merten@uni-bielefeld.de (M.M.); asaria.vogel@uni-bielefeld.de (A.V.); 3Thin Films & Physics of Nanostructures, Universitätsstrasse 25, 33615 Bielefeld, Germany; nfokin@physik.uni-bielefeld.de (N.F.); abecker@physik.uni-bielefeld.de (A.B.); bbueker@physik.uni-bielefeld.de (B.B.); andreas.huetten@uni-bielefeld.de (A.H.); 4Department of Regenerative Musculoskeletal Medicine, Institute for Musculoskeletal Medicine, University Hospital Muenster, Westfaelische Wilhelms University Muenster, Albert-Schweitzer-Campus 1, Building D3, 48149 Muenster, Germany; daniel.kronenberg@ukmuenster.de (D.K.); Richard.Stange@ukmuenster.de (R.S.); 5Protestant Hospital of Bethel Foundation, Department of Diagnostic and Interventional Radiology, Burgsteig 13, 33617 Bielefeld, Germany; guenther.wittenberg@evkb.de; 6Department of Cell Biology, Bielefeld University, Universitätsstrasse 25, 33615 Bielefeld, Germanyc.kaltschmidt@uni-bielefeld.de (C.K.); 7Bielefeld Institute for Nanoscience (BINAS), Bielefeld University, Universitätsstrasse 25, 33615 Bielefeld, Germany

**Keywords:** bone regeneration, collagen, nanopores, micropores, Spongostan

## Abstract

Scaffold materials for bone regeneration are crucial for supporting endogenous healing after accidents, infections, or tumor resection. Although beneficial impacts of microtopological or nanotopological cues in scaffold topography are commonly acknowledged, less consideration is given to the interplay between the microscale and nanoscale. Here, micropores with a 60.66 ± 24.48 µm diameter ordered by closely packed collagen fibers are identified in pre-wetted Spongostan, a clinically-approved collagen sponge. On a nanoscale level, a corrugated surface of the collagen sponge is observable, leading to the presence of 32.97 ± 1.41 nm pores. This distinct micro- and nanotopography is shown to be solely sufficient for guiding osteogenic differentiation of human stem cells in vitro. Transplantation of Spongostan into a critical-size calvarial rat bone defect further leads to fast regeneration of the lesion. However, masking the micro- and nanotopographical cues using SiO_2_ nanoparticles prevents bone regeneration in vivo. Therefore, we demonstrate that the identified micropores allow migration of stem cells, which are further driven towards osteogenic differentiation by scaffold nanotopography. The present findings emphasize the necessity of considering both micro- and nanotopographical cues to guide intramembranous ossification, and might provide an optimal cell- and growth-factor-free scaffold for bone regeneration in clinical settings.

## 1. Introduction

Bone regeneration is a highly regulated process that is crucial for endogenous healing after accidents, infections, or tumor resection. Disorders of bone healing (e.g., non-union or large bone defects) still represent major challenges in clinical care to date [1,2]. These require consequently additional surgeries and complex post-operative treatments for patients, which burden the health systems with relevant costs [3]. Although autologous bone is still optimal for bone replacement, donor site morbidity and limited available amounts are detrimental to autologous bone grafting, emphasizing the necessity for applying bone substitute materials [4,5]. However, bone tissue engineering by substitute materials is a complex and dynamic process, since natural bone combines a unique nano- and microtopography, allowing vascularization as well as migration and differentiation of stem cells and osteogenic progenitor cells, which deposit calcium to finally form the bone [6]. In particular, within the microscale of the bone, repeating osteon units comprising layers of collagen fibers facilitate a central canal containing blood vessels and nerves [6]. On a nanoscale level, triple-helix collagen molecules assembled to microfibrils show characteristic single D periodicity measuring 67 ± 5 nm and a gap region of approximately 30 nm length [7,8,9,10,11]. Providing the natural scaffold for apatite crystals, this nanotopological feature is crucial for bone formation [12,13]. Extending these findings, we recently identified a novel nanotopography with pores measuring 31.93 ± 0.97 nm in diameter on the surface of rat collagen type I fibers and demonstrated these pores to be sufficient to induce osteogenic differentiation of adult human stem cells in vitro [14]. Transferring these observations to nanotechnological and medical applications, we successfully produced a nanocomposite with 34 ± 14 nm pores by using a thermally induced crosslinking reaction of self-assembled SiO_2_ nanoparticles with oleic acid molecules, which likewise guided osteogenic differentiation of human stem cells in vitro [14]. These findings are in line with several studies demonstrating that changes in the nanotopography of artificial surfaces directly regulate differentiation of stem cells [6,15,16,17]. On a microscale level, Petersen and colleagues recently showed the importance of scaffold microarchitecture for secondary bone healing. Here, implantation of an engineered porcine collagen scaffold with a channel-like pore architecture comprising micropores of 89 ± 15 μm significantly improved fracture healing via endochondral ossification in a critical-size model in a rat femur [18].

In the present study, we took advantage of this knowledge by identifying the optimal nano- and microtopography for intramembranous ossification-driven bone regeneration in the cell- and growth-factor-free material Spongostan. According to its initial description by Bing, Spongostan is composed of hardened gelatin comprising formalin (1.5%) and lauryl alcohol (2%) [19]. Commonly used as a hemostatic collagen sponge in a broad range of clinical settings, Spongostan is a longstanding Food and Drug Administration-approved material, which is inexpensive, biocompatible, and non-allergenic [20]. The suitability of Spongostan as a carrier of stem cells or growth factors for bone regeneration was already described in vivo [21,22], and Spongostan was also shown to allow migration and osteogenic differentiation of preosteoblasts exposed to osteoinductive biochemical cues [23]. However, a direct guidance of bone regeneration solely by micro- and nanotopographical cues present on Spongostan has not been demonstrated so far. Here, we identify the presence of micropores with 60.66 ± 24.48 µm diameter and nanopores of 32.97 ± 1.41 nm size in wet porcine Spongostan, and demonstrate that this distinct micro- and nanotopography is sufficient for functional regeneration of a critical-size calvarial rat bone defect. While providing an optimal scaffold for bone regeneration in clinical settings, our data, therefore, emphasize the necessity of considering both the microtopography and nanotopography of scaffolds for guiding intramembranous ossification and bone recovery.

## 2. Materials and Methods

### 2.1. Study Design

The study design is depicted in Figure 1. Briefly, micropores and nanopores were identified in Spongostan, followed by assessment of their osteoinductive capacity in vitro. For investigation of bone regeneration in vivo, Spongostan was transplanted into critical-size calvarial defects. Next to an empty control, we applied sole collagen fibers (control lacking the microtopography of Spongostan) and Spongostan masked with nanoparticles (control lacking nano- and microtopography).

### 2.2. Spongostan

Spongostan (Ferrosan Medical Devices, Søborg, Denmark; marketed by Ethicon Biosurgery, Johnson and Johnson, New Brunswick, NJ, United States) was commercially purchased, cut into cubic (1 mm^3^, for in vivo experiments) or disc shapes (5 mm diameter × 5 mm height, for in vitro experiments), and pre-wetted in 0.9% NaCl solution (B. Braun Melsungen AG, Melsungen, Germany) for 30 min prior to transplantation.

### 2.3. Collagen Fibrils

Collagen fibrils were isolated from rat tail tendons, as previously described [14]. Briefly, tendons were cut into small pieces and incubated for 2 h at 37 °C in laundry detergent (Persil Megaperls; Henkel AG, Düsseldorf, Germany), followed by thorough washing and decomposition into fibrils using a liquid nitrogen-cooled mortar and subsequent ultrasonic disintegration. Pure collagen fibrils were collected by centrifugation and checked by high-power phase contrast microscopy prior to transplantation. Preparation of collagen fibrils for scanning electron microscopy (SEM) was done, as previously described [14].

### 2.4. SiO2 Nanoparticles and Coating of Spongostan

We synthesized SiO_2_ nanoparticles measuring 146 nm in diameter, following the protocol of Stöber [24] and our previously described protocol [14], while omitting the arrangement of nanoparticles to a closely packed 2D superstructure. Spongostan, which was pre-wetted as described above, was coated by rolling in SiO_2_ nanoparticles prior to transplantation.

### 2.5. Light Microscopy

Light microscopic imaging of dry Spongostan was done using the Keyence VHX microscope (Keyence, Osaka, Japan) without prior embedding. For fluorescence microscopy, dry and pre-wetted cubic sponges were embedded in Mowiol (laboratory-made) and imaged using confocal scanning microscopy (LSM 780, Carl Zeiss, Jena, Germany) and corresponding ZEN software.

### 2.6. Scanning Electron Microscopy

Very small amounts of the freeze-dried collagen fibers, Spongostan, or Spongostan coated with SiO_2_ nanoparticles were deposited onto a glass substrate and coated with a 4-nm thick ruthenium layer by magneton sputtering to prevent electron charging on the surface. The best resolution was obtained when working with an acceleration voltage between 2 and 3 kV, resulting in an electron current of 0.34 nA. For imaging of structural features of the SiO_2_ nanocomposite, a voltage of 5–10 kV was used (Field Electron and Ion Company, FEI, Hillsboro, OR, United States).

### 2.7. Osteogenic Differentiation of Human Stem Cells in Spongostan and Detection of Calcium Deposition

Human inferior turbinates were obtained during routine surgery after an informed consent process, according to local and international guidelines (Bezirksregierung Detmold, Germany). Inferior turbinate stem cells (ITSCs) were isolated and cultivated, as previously described [25,26]. Isolation and experimental procedures were ethically approved by the ethics commission of the Ärztekammer Westfalen-Lippe and the medical faculty of the Westfälische Wilhems-Universität (approval reference number 2012-15-fS, Münster, Germany). Mesenchymal Stem Cells (MSCs) obtained from bone marrow of healthy donors were commercially purchased (Lonza Group Ltd., Basel, Switzerland) and cultivated in Dulbecco’s Modified Eagle’s Medium (DMEM) supplemented with 10% fetal calf serum. For MSCs, on-site patient consent and approval from the Ethics Committee were not needed, since they were commercially available. For osteogenic differentiation in Spongostan, cells were seeded at a density of 10^4^ cells/sponge onto the gelatin sponges cut into disc shapes (5 mm diameter, 5 mm height) and cultivated in DMEM with 10% fetal calf serum (MSCs), or DMEM F12 supplemented with 10% human blood plasma, 40 ng/mL Fibroblast Growth Factor (FGF)-2, 20 ng/mL Epidermal Growth Factor (EGF), and B27 supplement (ITSCs; see [26]). For induced osteogenic differentiation, cells seeded on the sponges were exposed to osteogenic induction medium (OIM), containing DMEM supplemented with 10% fetal calf serum, 100 nM dexamethasone, 10 nM β-glycerophosphate, and 0.05 mM L-ascorbic acid-2-phosphate, according to our previous studies [25,26]. Calcium deposition was visualized after 7, 14, 21, and 28 days by Alizarin Red S staining, as we previously described [14]. Alizarin Red stainings were quantified using photometric measurement according to [27].

### 2.8. Immunocytochemistry

Cultivation of ITSCs in Spongostan as described above was followed by washing the sponge with phosphate buffered saline (PBS) and subsequent preparation for cryosectioning by covering with Tissue Tek (Sakura Finetek Europe B.V., Alphen aan den Rijn, Netherlands) and freezing in −30 °C 2-methylbutane. Cryosections were fixed in 4% paraformaldehyde for 15 min at RT and permeabilized with 0.02% Triton X-100 (Sigma-Aldrich, Taufkirchen, Germany), followed by incubation with primary α-human Osterix/Sp7 antibody (10 µg/mL, R&D Systems, Minneapolis, MN, USA) for 1 h at room temperature (RT). Secondary antibody conjugated with Alexa-555 (Life Technologies, Eugene, OR, USA) was diluted to 1:300 and incubated for 1h at RT under exclusion of light. Nuclear counterstaining was performed with DAPI (4′,6-diamidino-2-phenylindole) (AppliChem, Darmstadt, Germany). Cells were embedded in Mowiol and imaged using a LSM 780 (Carl Zeiss, Jena, Germany) and corresponding ZEN software (Zen 2011 sp7 fp3 (black) 64 bit, Carl Zeiss, Jena, Germany).

### 2.9. Animals

This study was approved by the responsible authority (LANUV, Approval Ref. No. 81-02.04.2018.A188). All animal work was performed in accordance with the policies and procedures established by the Animal Welfare act, the National Institutes of Health Guide for Care and Use of Laboratory Animals, and the National Animal Welfare Guidelines.

For this study, 20 male white Wistar rats with an average weight of 300 g at the day of surgery and an age of about 8 weeks were randomly assigned to either group A (empty control, n = 5), group B (collagen fibers, n = 5), group C (Spongostan, n = 5), or group D (Spongostan masked with SiO_2_ nanoparticles, n = 5). All animals are from our own breeding at the University of Bielefeld and were housed 2-per-cage under a 12h light/dark cycle with food and water provided *ad libitum*. A minimum number of rats was used and all efforts were made to minimize potential suffering in accordance to the 3R (replacement, reduction, refinement) guidelines.

### 2.10. Animal Surgery

The rats were administered an intraperitoneal injection of ketamine hydrochloride (60 mg/kg, Inresa Arzneimittel GmbH, Freiburg, Germany) and medetomidine (0,3 mg/kg, Dechra Veterinary Products Ltd., Shrewsbury, United Kingdom) after subcutaneous injection of tramadol (20 mg/kg, Grünenthal GmbH, Aachen, Germany). The surgical procedure followed the description of Spicer and colleagues [28]. After shaving the head from the bridge of the snout between the eyes to the caudal end of the skull using an electric clipper, the animal was transferred to the operating table on a heating pad (37 °C). An iodine swab was used to clean the area of surgery. Eye ointment (Bepanthen Augensalbe, Bayer AG, Berlin, Germany) was used for eye protection. An approximately 2 cm skin incision was made midline over the scalp from occiput to the base of the nasal bone using a scalpel. After preparation of the subcutaneous tissue, the periosteum was sharply divided at the sagittal midline, pushed laterally, and elevated from the bone. To protect the periosteum and the skin, sutures (Vicryl Plus 3-0, Ethicon, Johnson and Johnson, New Brunswick, NJ, USA) were placed through the periosteum to apply lateral traction. Afterwards, the trepanation was performed on both sides using a 5 mm hollow drill (Trephines 229 RAL 040: Hager and Meisinger GmbH, Neuss, Germany; Implantmed: W&H Dentalwerk Bürmoos GmbH, Bürmoos, Austria) on the left side first, followed by the right side. Trepanation was performed at a constant speed of 2000 rpm under continuous irrigation using dropwise sterile saline solution. At regular intervals, the drill was raised to check the depth. As soon as the remaining bone became translucent, its thickness was additionally assed by gently pressing with the elevator. Trepanation using the drill was stopped when a near full thickness cut through the calvarium was reached. The calvarium was removed using the elevator. Then, the blade of the elevator was placed into the defect margin and moved circumferentially around the defect. Afterwards, the circular bone piece was lifted out. The defect was washed using sterile saline solution, and depending on the group assigned to the animal, the holes were treated accordingly (empty, collagen fibers, Spongostan, or Spongostan masked with SiO_2_ nanoparticles). Thereafter, the periosteum was closed with sutures (Prolene 5-0, Ethicon, Johnson and Johnson, New Brunswick, NJ, USA). The skin was closed by four single back-and-forth sutures (Prolene 3-0, Ethicon, Johnson and Johnson, New Brunswick, NJ, USA).

### 2.11. Postoperative Care

Postoperatively, the heads of the rats were cleaned and animals were placed in a warmed incubator. They were housed singly for 3 days. For further analgesia, tramadol was added to the water bottles for five days (Grünenthal GmbH, Aachen, Germany, 2.5 mg/100mL drinking water, sweetened with 5% sugar).

### 2.12. Euthanasia and Sample Extraction

Thirty days postoperatively, animals were euthanized using the Exposure Line carbon dioxide box bioscape (Ehret, Freiburg, Germany). Thereafter, a section of the calvarium was extracted by removing soft tissue and surrounding bone. Specimens were fixated and stored using 4% paraformaldehyde (PFA).

### 2.13. Micro-CT

All specimens underwent micro-computed tomography (µCT) using a SkySkan 1176 (Bruker, Kontich, Belgium). Scans were performed at an isotropic resolution of 8.9 µm, with 50 kV energy at 500 µA intensity, and with an angle shift of 0.5° per image. To reduce artefacts, five pictures per angle were averaged. Pictures underwent axial reconstruction for further evaluation. The skulls were always aligned in the same manner, allowing transversal slices to be appreciated. To determine the volume of new bone formation, a circular region of interest (ROI) was placed into each defect. The volume of interest (VOI) was set by an upper and lower layer in the Z-direction, which was defined as the first appearance of bone on the corresponding slice. After the definition of the VOI, a semi-automated protocol allowed the standardized quantification of total volume, bone volume, and bone mineral density (BMD) of the newly formed bone inside the VOI. For BMD evaluation, the scanner was calibrated using a two-density phantom with hydroxyapatite at 0.25 and 0.75 g/cm^3^.

### 2.14. Statistics

Graph pad prism software was applied for statistical evaluation of measured bone volumes in individual defects (n = 6). Here, ***P <* 0.01 was considered significant using one-tailed Mann–Whitney test with a 95% confidence interval (CI). For statistical evaluation of quantified Alizarin Red concentrations, **P <* 0.05 was considered significant using one-tailed Mann–Whitney test with a 95% CI.

### 2.15. Histology

Specimens fixed with 4% of paraformaldehyde (PFA) were decalcified using ethylenediaminetetraacetic acid (EDTA) and embedded in paraffin. Sectioned specimens were stained via trichrome staining according to Goldner, while nuclei were stained with hematein followed by microscopical examination.

## 3. Results

### 3.1. Identification of Micropores of 60.66 ± 24.48 µm Diameter in the Clinically Approved Collagen Sponge Spongostan

We determined the topography of the collagen sponge Spongostan, which is FDA-approved and commonly applied as a hemostatic sponge in a broad range of clinical settings [20]. Using light microscopy, we identified the presence of micropores in dry Spongostan (Figure 2A–B). Further characterization of this microtopological feature of dry Spongostan by confocal laser scanning microscopy revealed micropores of 130.52 ± 42.15 µm diameter (Figure 2C).

After allowing the collagen sponge to pre-wet for 30 min followed by confocal laser scanning microscopy, we observed a reduction of the diameter of micropores to 60.66 ± 24.48 µm (Figure 2D). Extending the findings by Petersen and coworkers [18] and in accordance with the suitability of Spongostan as a carrier of stem cells or growth factors for bone growth [21,22], we suggested the here-identified microtopographical feature of Spongostan to be beneficial for bone regeneration.

### 3.2. Spongostan Reveals a Distinct Nanotopography of 32.97 ± 1.41 nm Pores

In addition to its microtopography, we aimed to identify distinct nanotopographical features of Spongostan. On the surface of native collagen type I fibers, we recently identified pores measuring 31.93 ± 0.97 nm, which were sufficient to induce osteogenic differentiation of adult human stem cells [14] (Figure 3A). Within the marked rectangular area of the Spongostan membrane (Figure 3B), a typical corrugated surface is always resolved in scanning electron microscopy (SEM) micrographs, as seen in Figure 3C. As with the native collagen type I fibers, we observed a nanoporous, corrugated surface on Spongostan using SEM (Figure 3C). Notably, the corrugated surface of Spongostan can be represented as a model assuming a tight packing of collagen fibers (Figure 3D). We also incorporated respective areas of single D repeats (Figure 3D, green areas) and gap regions (Figure 3D grey) in our proposed model of collagen filament ordering in Spongostan. In particular, we believe the microstructure of Spongostan to be composed of microsized membrane-like cells, which are locally highly ordered by closely packed collagen fibers (Figure 3B–D).

In order to extract microstructural features of Spongostan, the SEM image contrasts were quantitatively evaluated, employing the autocorrelation R(K) of normalized intensity profiles, which were measured directly from the SEM images (Figure 3E) using the following equation:$$R(K)=\left[\frac{N}{N-K}\right]\frac{\sum_{i=1}^{N-K}\left(I_{i}-I_{0}\right)\left(I_{i+K}-I_{0}\right)}{\sum_{i=1}^{N}{\left(I_{i}-I_{0}\right)}^{2}}.$$
where K is the length of an intensity interval, which is compared to the whole intensity profile and is taken at N positions. I_0_ is the average intensity and I_i_ is the average intensity at position i. The prefactor [N/(N−K)] was considered to compensate for the different numbers of summation ranges. For evenly spaced objects, the first minimum of R(K)_min_ is the measure of the mean size of these objects present in the total intensity profile. The first maximum of R(K)_max_ indicates the mean separation distance between these objects. Distinct microstructural features could be extracted from the measured R(K)_min_- and R(K)_max_ – values, particularly resulting in the observation of a distinct nanotopography comprising 32.97 ± 1.41 nm pores (Figure 3F). Notably, the nanotopographical feature observed here in Spongostan, as well as on native collagen fibers, was already shown to be sufficient to induce osteogenic differentiation of adult human stem cells [14]. Thus, we suggested the here-observed nanotopography of Spongostan to be likewise beneficial for osteogenic differentiation and bone regeneration in vivo and in vitro.

### 3.3. Adult Human Stem Cells Efficiently Undergo Osteogenic Differentiation by the Micro- and Nanotopography Present on Spongostan

To assess potential osteoinductive effects of Spongostan based on its nano- and microtopography in vitro, we seeded adult human neural crest-derived stem cells (NCSCs) isolated from the inferior turbinate (inferior turbinate stem cells, ITSCs) [25] and adult human mesenchymal stem cells (MSCs) on the collagen sponge. While MSCs are drivers of femoral bone regeneration [29], ITSCs also show a great capability for osteogenic differentiation [30,31], thus representing a highly promising cellular model system for bone recovery within the craniofacial region. After 7 days of culture with Spongostan, ITSCs and MSCs showed first signs of osteogenic differentiation, indicated by a slight Alizarin Red S-stained calcium deposition (Figure 4A). Further cultivation of ITSCs and MSCs in Spongostan for up to 28 days resulted in strong mineralization within the scaffold structure, shown by Alizarin Red S-staining, demonstrating successful osteogenic differentiation of adult human stem cells solely triggered by the nano- and microtopography of Spongostan (Figure 4A, C). Additional exposure of MSCs and ITSCs cultivated in Spongostan to an osteogenic induction medium (OIM) [26,31] likewise resulted in a strongly enhanced Alizarin Red S-stained calcium deposition, especially for MSCs (Figure 4B). In particular, we observed an increased osteoinductive effect of Spongostan and OIM compared to sole Spongostan for MSCs (Figure 4C), suggesting the addition of OIM to more closely mimic endogenous bone regeneration in vivo. Differentiation of MSCs driven by Spongostan with additional OIM for 28 days led to a synergistic effect, resulting in more pronounced calcium deposition compared to ITSCs (Figure 4C). On the contrary, no signs of osteogenic differentiation were observable in the negative controls (Figure 4A-B, cells in Spongostan at d0; Figure 4D, cell-free Spongostan). Pure Spongostan showed micropores defined by locally highly ordered and closely packed collagen fibers (Figure 3B-D; Figure 4D, arrowheads). In contrast to cell-free Spongostan, seeding of ITSCs resulted in migration into the sponge, followed by differentiation into Osterix-positive osteoblasts after 21 days (Figure 4D, arrows). Notably, ITSC-derived osteoblasts (Figure 4D, arrows) were localized directly at the border of the micropores in direct contact with closely packed collagen fibers (Figure 4D, arrowheads), showing the distinct nanotopography of 32.97 ± 1.41 nm pores (see also Figure 3B-F). Therefore, we demonstrate that the identified micropores allow migration of stem cells, which are further driven towards osteogenic differentiation by scaffold nanotopography.

### 3.4. The Micro- and Nanotopography of Spongostan is Solely Sufficient to Completely Regenerate a Critical-size Calvarial Rat Bone Defect

We next determined the suitability of the identified distinct micro- and nanotopography of Spongostan for regeneration of a critical-size calvarial rat bone defect in vivo. A critical-size calvarial bone defect is defined as the smallest intraosseous wound in a bone that will not heal spontaneously [32]. Trepanation using a 5 mm hollow drill was followed by transplantation of sole collagen fibers comprising 31.93 ± 0.97 nm pores, or Spongostan combining this nanotopographical feature (32.97 ± 1.41 nm pores) with micropores of 60.66 ± 24.48 µm diameter (Figure 5A). While no bone regeneration was observable via micro-computed tomography (µCT) in the control (Figure 5B-C), application of sole collagen fibers alone already led to partial but significant closure of the critical-size defect (Figure 5B, D). Notably, transplantation of Spongostan into the critical-size calvarial bone defect resulted in its complete closure (Figure 5E) and a significantly elevated bone volume compared to control and even to sole collagen fibers (Figure 5B). Histological examination of the newly formed bone confirmed this increase in bone tissue volume 4 weeks after transplantation of Spongostan in comparison to control and sole collagen fibers (Figure 5F). We also observed a more sponge-like appearance of the new bone tissue in the Spongostan-transplanted animals compared to collagen fibers, suggesting the whole collagen sponge to undergo ossification (Figure 5F). Importantly, and as a further quality control, we did not observe any unmineralized connective tissue in the lesion after transplantation of Spongostan (Figure 5F). Thus, we demonstrate here that the distinct micro- and nanotopography combining 60.66 ± 24.48 µm pores and 32.97 ± 1.41 nm Spongostan pores is sufficient for complete regeneration of a critical-size calvarial rat bone defect.

### 3.5. Successful Masking of Micro- and Nanopores Using SiO2 Nanoparticles Impairs the Osteoinductive Properties of Spongostan In Vivo

To further validate whether the effects of Spongostan on bone regeneration are driven by its nano- and microtopography, we aimed to mask the identified topographical features of Spongostan by utilizing SiO_2_ nanoparticles. In a previous study, we produced SiO_2_ nanocomposites by thermally cross-linking SiO_2_ nanoparticles, which were characterized by a diameter of 146 ± 36 nm with resulting pore sizes of 34 ± 14 nm in the nanocomposites when three particles touched each other in a close packed 2D particle monolayer arrangement [14]. Here, the intention for these nanoparticles is somewhat differently focused, as we manually applied nanoparticles of 146 nm diameter in thick 3D layers to destroy the crystallographic order of the 2D monolayers. Manual coating of Spongostan with 146 nm SiO_2_ nanoparticles (Figure 6A) led to masking of its microporous surface (Figure 6B).

In comparison to uncoated Spongostan (Figure 6C), SEM revealed the presence of 146 nm SiO_2_ nanoparticles on the surface of the collagen sponge after nanoparticle coating (Figure 6D). SiO_2_ nanoparticles were observable as unordered 3D layers on the surface of coated Spongostan, thus masking its distinct nanotopography (Figure 6D) and additionally blocking migration of stem cells into the sponge. Notably, transplantation of nanoparticle-coated Spongostan into critical-size calvarial defects (Figure 7A) completely impaired bone regeneration, as observable by a strongly reduced bone volume compared to uncoated Spongostan (Figure 7B), with no signs of closure of the lesion (Figure 7C–D). These observations demonstrate that blocking the distinct micro- and nanotopography of Spongostan completely inhibits its capacity for calvarial bone regeneration in vivo.

## 4. Discussion

In the present study, we identified novel micro- and nanotopography in the clinically-approved collagen sponge Spongostan, which is solely sufficient to guide osteogenic differentiation of human stem cells in vitro and in vivo. At the microscale, Spongostan revealed micropores of 60.66 ± 24.48 µm diameter, which allow migration of adult stem cells into the sponge. In addition to this microtopological feature, Spongostan showed a distinct nanotopography, with 32.97 ± 1.41 nm pores guiding osteogenic differentiation of the stem cells. In line with this, we previously showed a similar nanotopography of 30 nm pores present within endogenous collagen type I fibers to be sufficient to induce osteogenic differentiation of adult human stem cells in vitro [14]. Notably, the nanotopographical feature observed here in Spongostan, as well as on native collagen fibers [14], showed a similar size as the gap region present in collagen microfibrils. Here, the assembly of triple-helix collagen molecules to microfibrils resulted in single D repeats of 67 ± 5 nm and a gap region of 30 nm [7,9,33]. The respective areas of single D repeats and gap regions were also incorporated in our proposed model of collagen filament ordering in Spongostan (see Figure 2D). In accordance with our previous observations [14], we likewise suggest a close relation between this gap region of single D repeats and the distinct nanotopography of 32.97 ± 1.41 nm pores present in Spongostan.

Demonstrating the osteoinductive capacity of Spongostan in vitro, our present findings show that cultivation of NCSCs and MSCs in Spongostan results in successful osteogenic differentiation. In line with this, Kuo and colleagues reported that Spongostan is biocompatible with murine preosteoblasts and facilitates their OIM-mediated differentiation into osteogenic deviates [23]. Electrospun gelatin was further shown as a suitable scaffold for osteogenic differentiation of human MG63 osteosarcoma cells by a defined differentiation medium, comprising β-glycerol phosphate, calcium chloride, and ascorbic acid [34]. Bone marrow-derived mouse MSCs were also reported to reveal signs of osteogenic differentiation after 10 days of cultivation in a gelatin-CaSO_4_-scaffold, suggesting the release of calcium ions from CaSO_4_ to trigger osteogenic differentiation [35]. Extending these promising findings, our present observations reveal that the distinct micro- and nanotopography present in Spongostan is solely sufficient for guiding adult human stem cells into osteogenic differentiation in vitro. In particular, the identified micropores allow migration of stem cells, which are further driven towards osteogenic differentiation by scaffold nanotopography. Of note, we observed the differentiation of MSCs by Spongostan with additional OIM to result in a more pronounced calcium deposition compared to ITSCs. With regard to this extraordinary broad differentiation capability and our own characterizations (data not shown), the commercially purchased MSCs used in the present study may be CD164^+^ skeletal stem cells, which were reported to highly efficiently undergo osteogenic differentiation [36].

Emphasizing the potential of our approach for bone regeneration in vivo, we next demonstrated that the distinct micro- and nanotopography of Spongostan is sufficient for complete regeneration of a critical-size calvarial rat bone defect (Figure 8A,B). The optimum size of micropores recommended for osteoconduction is mostly around or greater than 100 µm [37,38,39,40]. However, our findings give evidence that micropores with a 60.66 ± 24.48 µm diameter, and thus smaller than 100 µm, efficiently drive bone regeneration in vivo. Accordingly, channel-like micropores measuring 89 ± 15 μm in porcine collagen scaffold were observed to facilitate endochondral ossification in a critical-size defect of the rat femur [18]. Our present data not only extend these observations towards intramembranous ossification of the calvarial bone, but also emphasize the substantial impact of the interplay between scaffold micro- and nanotopography for bone regeneration. In this regard, transplantation of Spongostan led to significantly increased bone density in newly formed bone compared to the application of sole collagen fibers comprising 31.93 ± 0.97 nm pores [14] but lacking the microtopography of Spongostan. Therefore, we suggest the micro- and nanotopography of Spongostan combining 60 ± 24 µm pores and 30 nm pores to be superior to the sole nanotopographical feature of 30 nm pores present in collagen fibrils in terms of bone regeneration in vivo. In a comparable approach on the microscale level, Levengood and colleagues showed that calcium phosphate scaffold porosity of 5.31–4.07 µm combined with pores of 359 µm size allows migration of osteogenic cells and subsequent mineralization, leading to repair of a lesion in pig hemi-mandibles in vivo [41]. We suggest our present approach to facilitate migration of endogenous stem cells and osteogenic progenitor cells into the scaffold microscale architecture, while osteogenic differentiation is further guided by the nanotopography of 30 nm pores, as we recently demonstrated [14] (Figure 8A,B).

During calcium deposition initiated by the newly formed osteocytes and osteoblasts, we suggest a stepwise replacement of Spongostan by the new bone tissue, resulting in its complete replacement over time. However, we are not able to exclude the possibility of Spongostan being reabsorbed after transplantation, which represents a limitation of the present study. In addition, we did not assess potential inflammatory responses, since several studies already described that Spongostan does not trigger inflammatory responses after transplantation [20,21,23,42]. Emphasizing the impact of both micro- and nanotopography for bone regeneration, successful masking of micro- and nanopores using thick 3D layers of 146 nm SiO_2_ nanoparticles impaired the osteoinductive properties of Spongostan in vivo (Figure 8C). Here, we suggest the thick layers of nanoparticles to block the migration of stem cells into the Spongostan microstructure, while masking its nanotopography further impairs the osteoinductive capability of Spongostan (Figure 8C).

In addition to the interplay of micro- and nanostructures, our data emphasize the importance of nanotopography ordering in guiding osteogenic differentiation of human stem cells and bone regeneration. While unordered 3D layers of 146 nm SiO_2_ nanoparticles prevented bone regeneration in the present study, a close packaging of similar 146 nm SiO_2_ nanoparticles as a 2D monolayer arrangement leading to the presence of 34 ± 14 nm pores was shown to be osteoinductive [14]. In line with this, fate decisions of stem cells were reported to be directly influenced by distinct surface nanotopographies and their ordering [15,16,17]. Human MSCs were shown to undergo differentiation into osteoblast-like cells by cultivation on 70–100 nm titanium nanotubes, while exposure of MSCs to 30 nm tubes did not result in differentiation [43]. Huang and colleagues further revealed that the nanoscale order or disorder of an integrin-specific ligand has a crucial impact on osteoblast adhesion [44], highlighting the importance of nanotopographical ordering in bone growth and regeneration. Accordingly, we show that the combination of highly ordered 30 nm nanopores present in Spongostan, together with its distinct microtopography, drive calvarial bone regeneration in vivo.

By utilization of a clinical-approved collagen sponge [20], we further aimed to bridge the translational gap between “bench and bedside”. In addition to its clinical application as a hemostatic collagen sponge, Spongostan is frequently used as a carrier for stem cells or growth factors in bone recovery [21,22]. In particular, Spongostan was shown to sufficiently allow osteogenic differentiation of preosteoblasts via biochemical osteoinductive cues [23], and was likewise suggested as a carrier for stem cells in terms of iliac crest bone grafting [42]. Lin and colleagues demonstrated that gelatin sponge scaffolds are suitable for transplanting adipose-derived stem cells (ASCs) expressing bone morphogenetic protein 2 (BMP2) or transforming growth factor-β2 (TGF-β3) into critical-size calvarial defects in rabbits. Interestingly, ASCs comprising gelatin sponge scaffolds led to a more pronounced osteogenesis and calvarial healing compared to apatite-coated polylactide-co-glycolid (PLGA) scaffolds seeded with ASCs [22]. Likewise, Aquino-Martinez and coworkers reported a gelatin/CaSO4 scaffold loaded with MSCs from bone marrow to enhance bone formation in a critical-size calvarial defect in mice [35]. Arias-Gallo and colleagues utilized Spongostan as a cell-free carrier of fibroblast growth factor 1 (FGF-1) in a rat craniotomy model and achieved nearly complete bone regeneration [21]. In the present study, we extend these observations by demonstrating that the nano- and microtopography of Spongostan drives intramembranous ossification and calvarial bone regeneration in vivo, even in the absence of cells or growth factors.

## 5. Conclusions

In the present study, we identified novel micro- and nanotopological features of 60.66 ± 24.48 µm pores and 32.97 ± 1.41 nm pores on the collagen sponge Spongostan, which sufficiently guided osteogenic differentiation in vitro and bone regeneration in vivo. Accordingly, masking of its micro- and nanotopography using unordered 3D layers of SiO_2_ nanoparticles prevented bone regeneration. We conclude that the identified micropores allow migration of endogenous stem cells into Spongostan in vivo, whereby the cells are driven towards osteogenic differentiation by nanotopographical cues. Extending the broad range of studies focusing on nanotopography or microtopography of scaffolds, our findings, thus, emphasize the necessity to consider both microtopography and nanotopography of grafts to allow intramembranous ossification and bone recovery. As a future perspective, Spongostan may be directly applied in clinical settings of bone recovery. Here, translational gaps between bench and bedside may be circumvented, since Spongostan is already clinically and FDA-approved, and our approach does not rely on cells or growth factors as additives to improve regeneration.

## Figures and Tables

**Figure 1 cells-09-00654-f001:**
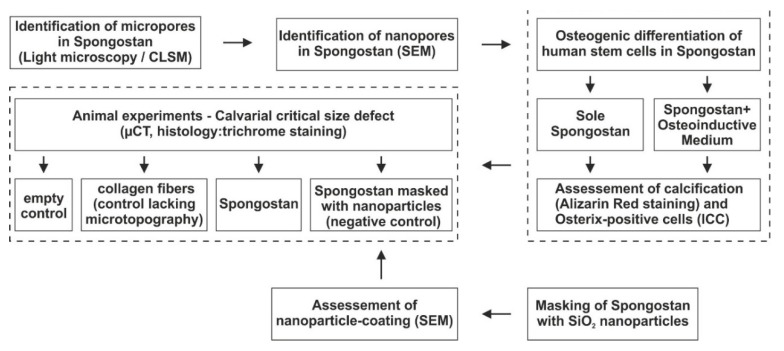
Flowchart of the study design. Briefly, micropores and nanopores were identified in Spongostan, followed by assessment of their capacity for osteoinduction in vitro and bone regeneration in vivo. CLSM, confocal laser scanning microscopy; SEM, scanning electron microscopy; ICC, immunocytochemistry; µCT, micro-computed tomography.

**Figure 2 cells-09-00654-f002:**
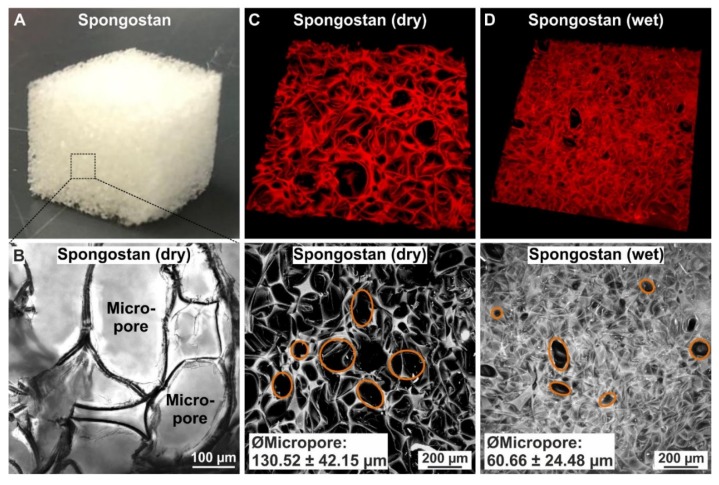
The collagen sponge Spongostan reveals micropores of 60.66 ± 24.48 µm diameter. (**A**–**B**) Macro- and light microscopical images of dry Spongostan showing the presence of micropores. (**C**–**D**) Confocal laser scanning microscopy utilizing the autofluorescence of Spongostan identified micropores of 130.52 ± 42.15 µm diameter in dry Spongostan, while pre-wetted Spongostan showed micropores of 60.66 ± 24.48 µm diameter.

**Figure 3 cells-09-00654-f003:**
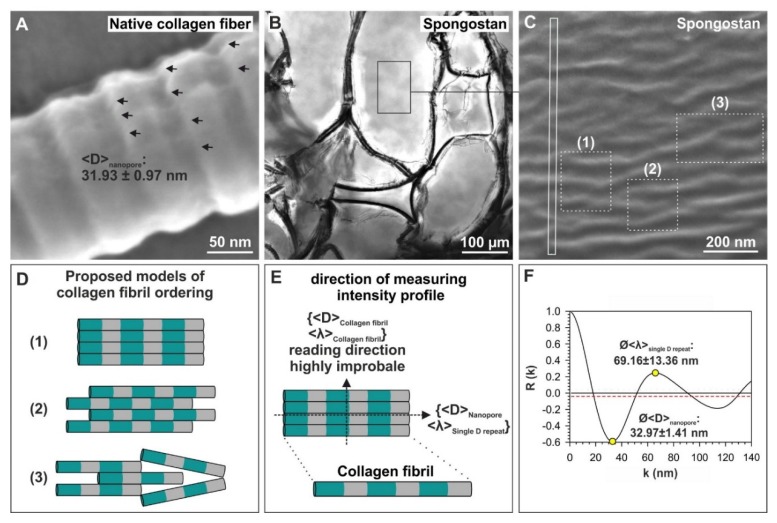
Identification of 32.97 ± 1.41 nm pores on the surface of Spongostan. (**A**) The surface of a native collagen type I fibril comprises pores with 31.93 ± 0.97 ± nm size [14]. (**B**) Light micrograph revealing a microstructural cell network of very thin membranes. (**C**) Scanning electron microscopy (SEM) micrograph of the area depicted in (B) showed the corrugated, nanoporous surface of Spongostan. (**D**) Representation of the corrugated surface of Spongostan as a model assuming a close packing of collagen fibers. Structures (1)–(3) observed in (C) are visualized as (1)–(3) in the model. (**E**–**F**) Analyses by autocorrelating the underlying intensity profile of SEM images resulted in R(K)_min_ (mean size of objects present in the total intensity profile) and R(K)_max_ (mean separation distance between these objects) values, which identify the distinct nanotopography of Spongostan, comprising 32.97 ± 1.41 nm pores. Equation and detailed definitions of R(K)_min_ and R(K)_max_ are included below.

**Figure 4 cells-09-00654-f004:**
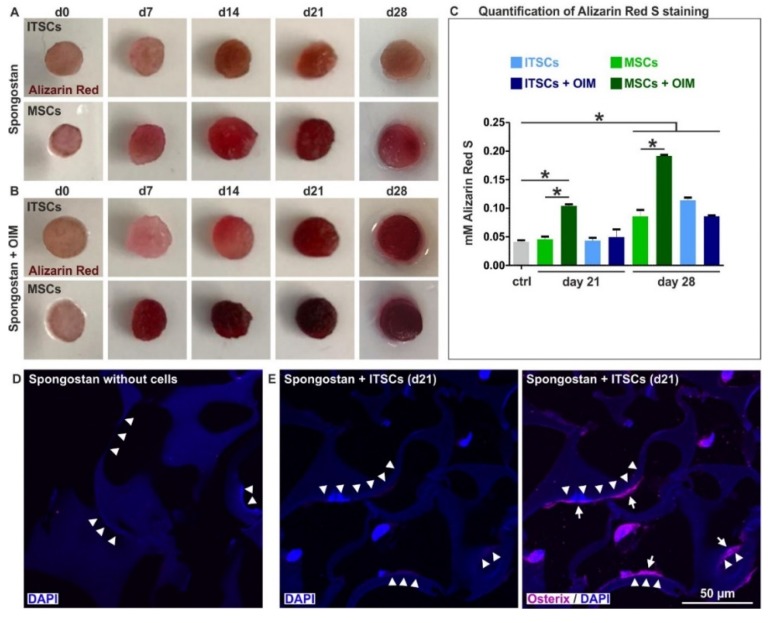
Osteogenic differentiation of human stem cells by the micro- and nanotopography present on Spongostan. (**A**) Adult human inferior turbinate stem cells (ITSCs) and adult human skeletal stem cells (MSCs) showed Alizarin Red S-positive calcium deposition after cultivation in the collagen sponge for up to 28 days, indicating successful osteogenic differentiation. (**B**) Additional exposure of MSCs and ITSCs cultivated in Spongostan to an osteogenic induction medium (OIM) resulted in increased Alizarin Red S-stained calcium deposition. (**C**) Quantification of Alizarin Red staining validated the observed osteoinductive effects of Spongostan, while also revealing an increased osteoinduction of Spongostan and OIM compared to sole Spongostan in MSCs. (**D**) Cell-free Spongostan showed a microstructural network of densly-packed collagen fibers (arrowheads), which form the micropores. (**E**) ITSCs successfully migrated into Spongostan followed by differentiation into Osterix-positive osteoblasts after 21 days (arrows), which were localized directly at the border of the micropores in direct contact with closely packed collagen fibers (arrowheads).

**Figure 5 cells-09-00654-f005:**
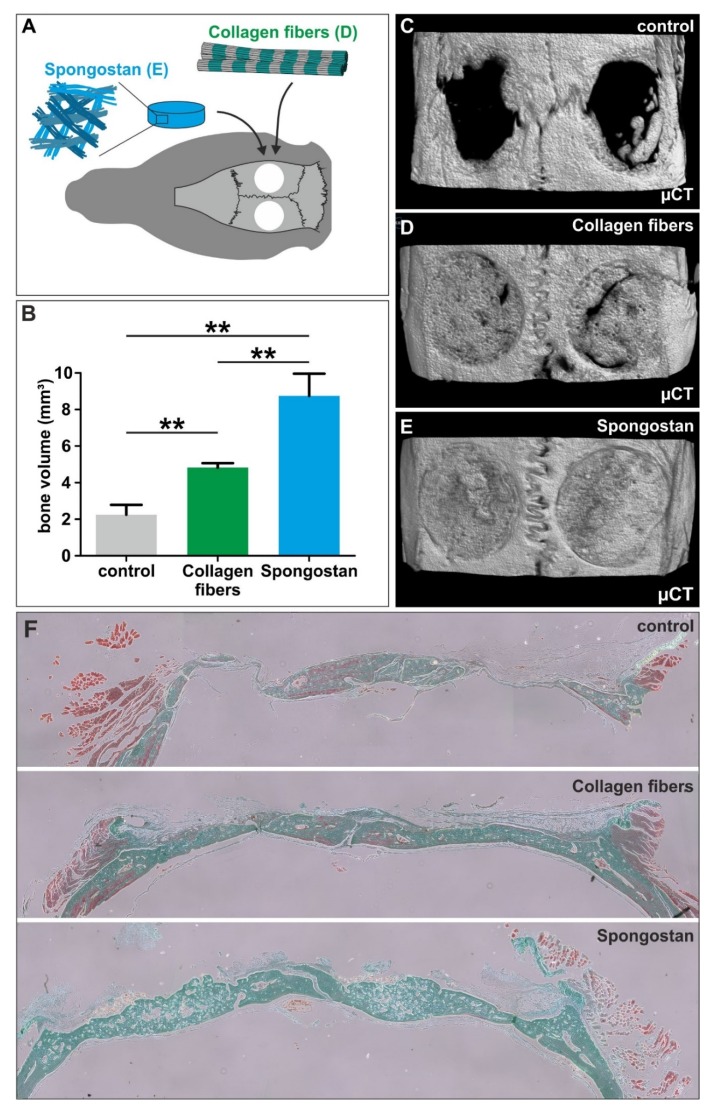
The micro- and nanotopography of Spongostan drives regeneration of a critical-size calvarial rat bone defect. (**A**) Schematic depiction of the transplantation strategy. (**B**) Analysis of bone volume demonstrated significantly increased bone regeneration 4 weeks after transplantation of Spongostan compared to sole collagen fibers or untreated control. (**C**–**E**) Representative micro-computed tomography (µCT) images of rat cranial defects 4 weeks after transplantation revealed a complete closure of the lesion in the Spongostan group, which was only in partially observable after transplantation of collagen fibers and was not observable in the control. (**F**) Histological examination of the newly formed bone using trichrome staining confirmed the increase in bone tissue volume 4 weeks after transplantation of Spongostan in comparison to control and sole collagen fibers. Bright green staining: mineralized bone tissue and collagen; light green staining: calcified cartilage matrix; red staining: osteoid; blue-black staining: nuclei; red-brown staining: cytoplasm; orange staining: erythrocytes.

**Figure 6 cells-09-00654-f006:**
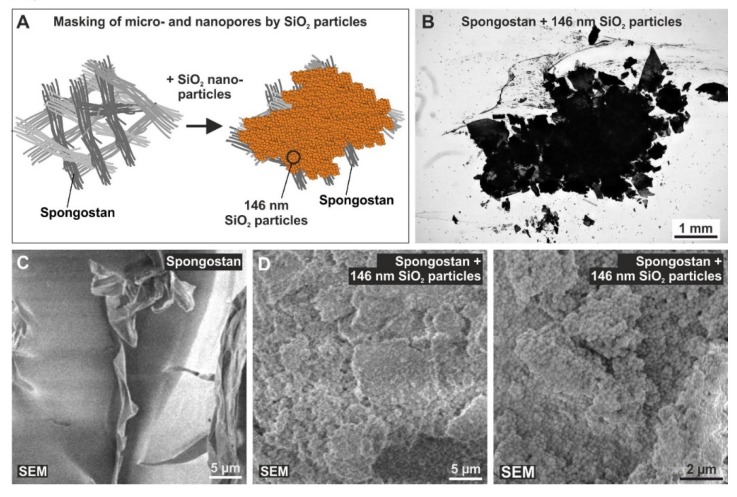
A successful masking of micro- and nanopores present on Spongostan using SiO2 nanoparticles. (**A**) Schematic depiction of the coating of Spongostan with 146 nm SiO_2_ nanoparticles to mask its micro- and nanotopography. (**B**) Manual coating of Spongostan with 146 nm SiO_2_ nanoparticles led to masking of its microporous surface. (**C**–**D**) Scanning electron microscopy showed the presence of 146 nm SiO_2_ nanoparticles as unordered 3D layers on the surface of collagen sponge after nanoparticle coating compared to uncoated Spongostan.

**Figure 7 cells-09-00654-f007:**
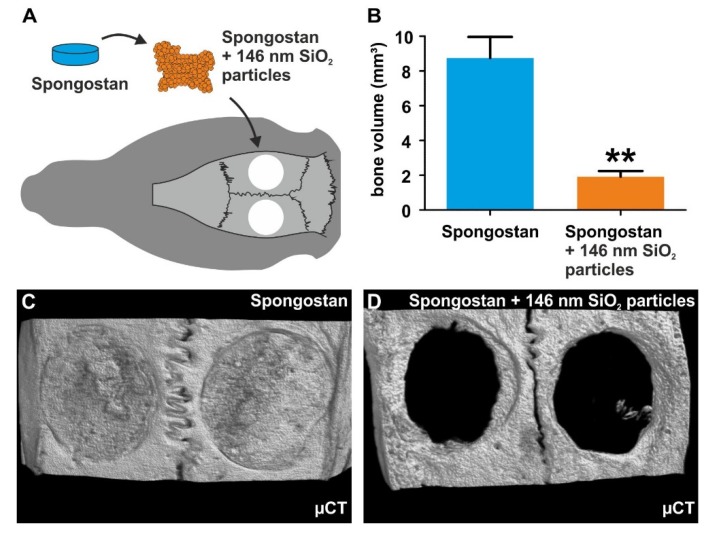
Masking of micro- and nanopores on Spongostan prevents bone regeneration in vivo. (**A**) Schematic depiction of the transplantation strategy. (**B**) Transplantation of nanoparticle-coated Spongostan into critical-size calvarial defects resulted in a significantly reduced bone volume compared to uncoated Spongostan. (**C**) Representative µCT images of rat cranial defects 4 weeks after transplantation showed no signs of closure of the lesion in the group, which received nanoparticle-coated Spongostan.

**Figure 8 cells-09-00654-f008:**
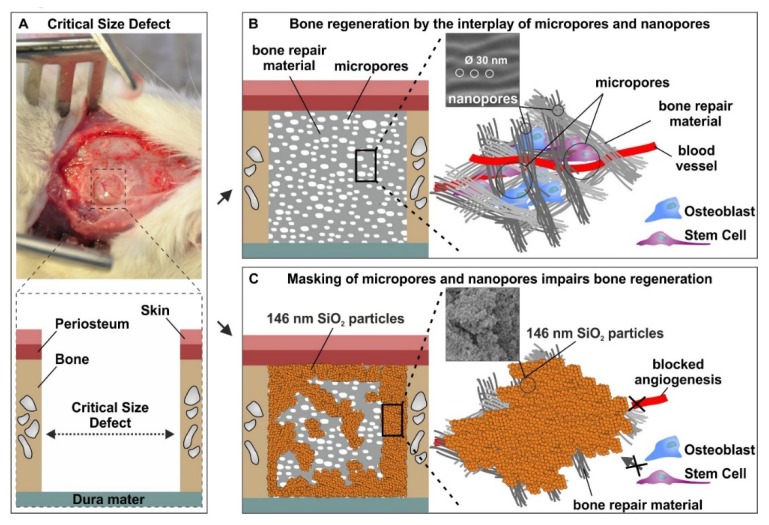
Bone regeneration by the interplay of micro- and nanotopography present on Spongostan. (**A**) Photographic and schematic illustration of the critical-size calvarial defect in rats. (**B**) Schematic depiction of our present approach, which allows migration of endogenous stem cells and osteogenic progenitor cells into the collagen scaffold microscale architecture, while osteogenic differentiation is further guided by the nanotopography of 30 nm pores. (**C**) Schematic depiction Spongostan coated with thick 3D layers of 146 nm SiO_2_ nanoparticles, which blocks migration of stem cells into the Spongostan microstructure, while masking its nanotopography further impairs the osteoinductive capability.

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
