# Peer review of "Bone Regeneration: A Novel Osteoinductive Function of Spongostan by the Interplay between Its Nano- and Microtopography"

_cells, 2020, doi:10.3390/cells9030654_

Round 1

Reviewer 1 Report

In the current manuscript, Vordemvenne et al. investigated the micro- and nano topography of spongostan. In a first step the authors analysed the topography and its effect on cells in both imaging and in-vitro experiments and afterwards corroborated their findings with a osseous defect rat-model. Bone regeneration remains a major challenge for orthopaedic surgeons. Despite numerous existing bone substitutes more studies are needed to improve the understanding of what determines successful and unsuccessful biomaterials. Thus, the purpose of the current study is of interest. In general, the study was well designed, the results are robust, the findings novel and supported by the data and the conclusion derives from the findings. All in all, this a exceptional study. 

I only have three minor suggestions/questions:

1) In my opinion, due to the numerous experiments, it would help to have a flowchart of the study design in the beginning of the MM section. This is more common in clinical studies, but it helps the reader to understand why each experiment was conducted and what follows next. 

2) Some studies found that SiO2 nanoparticles are toxic on cells. Do the authors think, that the complete lack of bone regeneration, derives only from the masked topography or could SiO2 as a material itself play a role? If they agree with me please include this into the limitations of the study. If not, please explain. 

3) It is unusual to include a figure in the conclusion, I would move that section into the discussion and keep the conclusion shorter. 

Author Response

1) In my opinion, due to the numerous experiments, it would help to have a flowchart of the study design in the beginning of the MM section. This is more common in clinical studies, but it helps the reader to understand why each experiment was conducted and what follows next. 

We thank the reviewer for raising this issue. We now included a flowchart of the study design (revised figure 1) as a separate paragraph in the materials and methods section (lines 88-100).

2) Some studies found that SiO2 nanoparticles are toxic on cells. Do the authors think, that the complete lack of bone regeneration, derives only from the masked topography or could SiO2 as a material itself play a role? If they agree with me please include this into the limitations of the study. If not, please explain. 

We thank the reviewer for raising this important question. From our point of view, the observed complete lack in bone regeneration derives only from the masked topography. In a previous study, we produced SiO2 nanocomposites by thermally cross-linking SiO2 nanoparticles with a diameter of 146 ± 36 nm (Greiner et al. Nanomedicine 2019, citation [14]). This cross-linking resulted in a SiO2 nanocomposite with pore sizes of 34 ± 14 nm when three particles touching each other in a close packed 2D particle monolayer arrangement. Notably, seeding of adult human stem cells on this SiO2 nanocomposite did not affect their survival but even resulted in osteogenic differentiation. On the contrary, unordered 3D layers of similar 146 nm SiO2 nanoparticles prevented bone regeneration in the present study. We thus conclude that nanotopography ordering of SiO2 nanoparticles is crucial for guiding or preventing bone regeneration and the applied SiO2 nanoparticles are non-toxic to endogenous stem cells. These aspects are discussed within the discussion sections (lines 480-486).

3) It is unusual to include a figure in the conclusion, I would move that section into the discussion and keep the conclusion shorter. 

We now moved the figure in the discussion section and shortened the conclusion accordingly.

Reviewer 2 Report

Very interesting manuscript on micro/nanotopography of Spongostan used in vivo as scaffold for bone regeneration.

The manuscript can be published in the present form

Author Response

We gratefully thank the reviewer for accepting our manuscript in its present form.

Reviewer 3 Report

the manuscript reports spongostan as a substrate for bone formation in vitro and in vivo. the manuscript is very well written and presented.

 Authors might want to provide a bit more info on the composition of the spongostan, if available.

Author Response

 Authors might want to provide a bit more info on the composition of the spongostan, if available.

We thank the reviewer for raising this issue. According to its initial description by Bing (Bing, 1947, Acta Pharmacol Toxicol (Copenh), 3 (4), 364-72), Spongostan is made of gelatin, formalin (1.5 %) and lauryl alcohol (2 %). Formalin and lauryl alcohol are added to fluid gelatin and the sponge is produced by beating air into the fluid. While formalin ensures hardening of the stiffening gelatin sponge, the surface-active compound lauryl alcohol allows to cut the dry sponge after production. We now included the composition of Spongostan together with the respective reference in introduction section (see lines 73-74).

Reviewer 4 Report

“Bone regeneration: A novel osteoinductive function of Spongostan by the interplay between its nano- and microtopography”

In tissue engineering research, the use of scaffolds plays a crucial role. They provide architecture and structural support while guiding tissue formation. Spongostan is a haemostatic and biocompatible gelatinous sponge used in surgery. This sponge has already proven to be an exceptional cell scaffold and template to guide tissue growth. However, Vordemvenne and colleagues aimed to describe the nano and microtopography of this sponge and demonstrate that the sponge’s topography is the responsible for bone regeneration in an environment free of growth factors. Although the novelty of the idea there are some inconsistencies that must be addressed before it is accepted for publication.

Major concerns

  • The authors are not able to demonstrate which topography of the sponge, nano or micro, is essential to be considered a good scaffold. The use of SiO2 nanoparticles masks both topographies found in the Spongostan. Therefore, the conclusion made by the authors that micropores allow migration of stem cells, which are further driven towards osteogenic differentiation by scaffold nanotopography is impossible to take. Please comment.
  • Spongostan is absorbable and although several studies have already described that it does not trigger a significant inflammatory response it should have been studied in the present study.
  • The native dispersed collagen fibers do not have the nano-scaffold structure of Spongostan (391-392). But at the same time acknowledge that the sole fibers of collagen lack the microtopography of Spongostan (lines 427-428). Please clarify.
  • Indeed, the Spongostan is growth-factor free however the authors used an osteogenic induction medium with dexamethasone, β-glycerophosphate and L-ascorbic acid-2-phosphatase which stimulate cell proliferation and induce osteoblast differentiation. Would the calcium deposition be the same if these supplements were not used?
  • Spongostan has been reabsorbed or totally replaced by bone. Please clarify.
  • The authors fail to describe the study’s limitations. Please include them.
  • A professional editor to review the manuscript’s grammar and syntax is not mandatory, but an exhaustive reading of the manuscript is important to correct some errors and to make clear the message they wish to convey to the reader, because sometimes it is necessary to reread much of what is behind to understand the text.

Minor concerns

Tramadol is written with lowercase and uppercase letters. Please uniform.

Lines 244-245. The authors state: “Notably, the corrugated surface of Spongostan can be modeled assuming a close packing of collagen fibers.” How can Spongostan can be modeled?

Figure 2A is the same as reference 14. Do the authors have permission to publish this image again?

The resolution of Figure 2 C should be improved. What do the authors mean by (1), (2) and (3) in this image?

Line 280: After defining an abbreviation, it should be used later (for example MSC, mesenchymal stem cells).

Line 322: Please correct Spongostan.

Line 326: Please, define µCT.

Line 328: Please, define “critical size” of calvarial bone defect.

Figure 4F: Please, provide the description of Masson’s Trichrome stain used in the present manuscript for the correct interpretation of the images.  

Line 428: Please correct “therefore”.

Authors contributions

Authorship is the collective responsibility of the authors, although it is my responsibility to allow you to appreciate the recommendations of the International Committee of Medical Journal Editors (ICMJE) on authorship criteria. Authors will meet four cumulative criteria, but none about “resources” contributions or “funding acquisition”. Collaborations are encouraged in every aspect of today’s life, but money should never be the reason for any collaboration.

Please define the contributions “investigation” and visualization”.

Please place funding in the funding section.

References section

Please, abbreviate the journal’s name of references 4; 6; 9; 10; 12; 13; 14; 17; 18; 22; 23; 30; 31, 37; and 40.

Reference 8: Please lowercase the K and the E from “BoneKEy”.

Author Response

Major concerns

The authors are not able to demonstrate which topography of the sponge, nano or micro, is essential to be considered a good scaffold. The use of SiO2 nanoparticles masks both topographies found in the Spongostan. Therefore, the conclusion made by the authors that micropores allow migration of stem cells, which are further driven towards osteogenic differentiation by scaffold nanotopography is impossible to take. Please comment.

We thank the reviewer for raising this issue. As pointed out by the reviewer, we applied SiO2 nanoparticles to mask both topographies found in Spongostan. Since masking of nano- and microtopographies completely impaired bone regeneration, we concluded that both topographies are crucial for bone recovery.  In figure 3E (revised figure 4E), we show that ITSCs successfully migrate into Spongostan, thus demonstrating the present micropores to allow migration of stem cells into the sponge. After migration and attachment to collagen fibers in the sponge, ITSCs differentiated into Osterix-positive osteoblasts after 21 day in vitro, in turn resulting in calcium deposition within the sponge (Fig. 3/ revised Fig. 4). Notably, ITSCs-derived Osteoblasts were localized directly at the border of the micropores and in direct contact to closely packed collagen fibers comprising the distinct nanotopography of 32.97 ± 1.41 nm pores (Fig. 2C-F/ revised Fig. 3C-F). This nanotopographical feature of Spongostan is similar to the nanoporous surface of native collagen type I fibers, where we recently identified pores with 31.93 ± 0.97 nm diameter. In this recent study, we showed that the 30 nm pores present on collagen type I fibers are sufficient to induce osteogenic differentiation of adult human stem cells (Greiner et al. Nanomedicine 2019, citation [14]). In this line, application of sole collagen fibers alone already led to partial but significant closure of the critical size defect in the present study. We thus concluded that Spongostan-mediated osteogenic differentiation is likewise driven by the similar scaffold nanotopography of 30 nm pores. In addition to our previous findings, this conclusion is based on the observation that adult stem cells differentiated in direct contact to collagen fibers with 32.97 ± 1.41 nm pores after migration into Spongostan (see above). As transplantation of Spongostan resulted in a significantly elevated bone volume compared to sole collagen fibers, we concluded the interplay between both micro- and nanotopography to be crucial for complete closure of the lesion. With regard to the observations discussed above, we summarized that micropores allow migration of stem cells, which are further driven towards osteogenic differentiation by scaffold nanotopography.

Spongostan is absorbable and although several studies have already described that it does not trigger a significant inflammatory response it should have been studied in the present study.

We thank the reviewer for this remark. As notified by the reviewer, several studies already demonstrated, that Spongostan does not trigger inflammatory responses in vivo (references 19, 20, 22 and 41). In this regard, we did not assess this aspect again in the present study. In addition, we are not able to address this matter on an experimental level due to the only very short time for addressing the reviewer’s comments (5 days). However, we now included this issue in the discussion section as a limitation of our study (see lines 472-474 and below).

Discussion section:

[…] In addition, we did not assess potential inflammatory responses, since several studies already described that Spongostan does not trigger inflammatory responses after transplantation [19,20,22,41]. Emphasizing the impact of both micro- and nanotopography for bone regeneration, successful masking of micro- and nanopores using thick 3D layers of 146 nm SiO2 nanoparticles impaired the osteoinductive properties of Spongostan in vivo (Fig. 8C). […]

The native dispersed collagen fibers do not have the nano-scaffold structure of Spongostan (391-392). But at the same time acknowledge that the sole fibers of collagen lack the microtopography of Spongostan (lines 427-428). Please clarify.

We thank the reviewer for this remark. Although Spongostan and native collagen fibers do not have a completely identical nano-scaffold structure, we observed a highly similar nanotopological feature in both materials. In particular, we identified 32.97 ± 1.41 nm pores as a distinct nanotopological feature of Spongostan, while the surface of native collagen type I fibers comprises 31.93 ± 0.97 ± nm pores. In the referred lines 391-392 (now lines 413-414), we compare the size of these nanotopographical features in Spongostan and on native collagen fibers and conclude that it is similar to the gap region present in collagen microfibrils. Although Spongostan and native collagen fibers share this similar nanotopological feature, native collagen fibers lack the microtopography of Spongostan, which we state in the respective lines 427-428 (now lines 449-450). The microtopography of Spongostan is defined by micropores with 60.66 ± 24.48 µm diameter, as we identified in the present manuscript. However, we did not observe these micropores in sole collagen fibers in the present or our previous study cells (Greiner et al. Nanomedicine 2019, citation [14]). Notably, transplantation of Spongostan led to significantly increased bone density of newly formed bone compared to the application of sole collagen fibers. We thus conclude the interplay between micro- and nanotopography of Spongostan to be superior to the sole nanotopographical feature of 30 nm pores present in collagen fibrils in terms of bone regeneration in vivo. We hope this discussion helped to clarify the remark raised by the reviewer.

Indeed, the Spongostan is growth-factor free however the authors used an osteogenic induction medium with dexamethasone, β-glycerophosphate and L-ascorbic acid-2-phosphatase which stimulate cell proliferation and induce osteoblast differentiation. Would the calcium deposition be the same if these supplements were not used?

We thank the reviewer for raising this question. We addressed this aspect in figure 3A (revised Fig. 4A), where we show that cultivation of ITSCs and MSCs in Spongostan without additional biochemical cues resulted in a strong mineralization within the scaffold structure. We observed an increased osteoinductive effect of Spongostan and OIM compared to sole Spongostan particularly for MSCs (Fig. 3B-C/ revised Fig. 4B-C), suggesting the addition of OIM to more closely mimic endogenous bone regeneration in vivo. However, our findings show that the nano- and microtopography of Spongostan is solely sufficient to trigger osteogenic differentiation of adult human stem cells in vitro, even in the absence of biochemical cues.

Spongostan has been reabsorbed or totally replaced by bone. Please clarify.

We thank the reviewer for raising this issue. After transplantation of Spongostan into the lesion, endogenous stem cells are suggested to migrate into the sponge, where they differentiate into osteoblasts and osteocytes. During calcium deposition initiated by the newly formed osteoblasts, we suggest a stepwise replacement of Spongostan by the new bone tissue. Over time, Spongostan may be further replaced by bone up to its complete replacement. However, as these aspects are only speculative, we did not include them in the discussion section so far but now listed them as limitations (see lines 469-472 and below).

Discussion section:

[…] During calcium deposition initiated by the newly formed osteocytes and osteoblasts, we suggest a stepwise replacement of Spongostan by the new bone tissue resulting in its complete replacement over time. However, we are not able to exclude the possibility of Spongostan being reabsorbed after transplantation, which represents a limitation of the present study. […]

The authors fail to describe the study’s limitations. Please include them.

We thank the reviewer for raising this issue. As discussed above we now included limitations of the study in the discussion section (see lines 469-474 and below).

[…] During calcium deposition initiated by the newly formed osteocytes and osteoblasts, we suggest a stepwise replacement of Spongostan by the new bone tissue resulting in its complete replacement over time. However, we are not able to exclude the possibility of Spongostan being reabsorbed after transplantation, which represents a limitation of the present study. In addition, we did not assess potential inflammatory responses, since several studies already described that Spongostan does not trigger inflammatory responses after transplantation [19,20,22,41]. […]

A professional editor to review the manuscript’s grammar and syntax is not mandatory, but an exhaustive reading of the manuscript is important to correct some errors and to make clear the message they wish to convey to the reader, because sometimes it is necessary to reread much of what is behind to understand the text.

We thank the reviewer for this remark. We now carefully reread the manuscript and corrected it in terms of clarity if necessary.

Minor concerns

Tramadol is written with lowercase and uppercase letters. Please uniform.

We corrected the spelling accordingly.

Lines 244-245. The authors state: “Notably, the corrugated surface of Spongostan can be modeled assuming a close packing of collagen fibers.” How can Spongostan can be modeled?

The sentence is replaced by: “Notably, the corrugated surface of Spongostan can be represented as a model assuming a tight packing of collagen fibers (Fig. 2D)”. We also changed the figure legend accordingly.

Figure 2A is the same as reference 14. Do the authors have permission to publish this image again?

Although Figure 2A shows the same observation as described in reference 14, a different picture is shown here. We nevertheless cited the respective reference now also in the figure legend. Since reference 14 is available under the Creative Commons CC-BY-NC-ND license, we are allowed to use of our work for non-commercial purposes.

The resolution of Figure 2 C should be improved. What do the authors mean by (1), (2) and (3) in this image?

We are not able to improve the resolution of Fig. 2C due to technical limiations. (1), (2) and (3) in Fig. 2 C represent the areas which were used for the model proposal in Figure 2 D, which is now clearly noted in the figure legend.

Line 280: After defining an abbreviation, it should be used later (for example MSC, mesenchymal stem cells).

We corrected the spelling of MSCs accordingly.

Line 322: Please correct Spongostan.

We corrected the spelling accordingly.

Line 326: Please, define μCT.

µCT is now defined by micro-computer tomography

Line 328: Please define “critical size” of calvarial bone defect

We now added a definition of a critical size defect in the results section (see lines 339-341).

Figure 4F: Please, provide the description of Masson’s Trichrome stain used in the present manuscript for the correct interpretation of the images.

The description of the staining is now added in the figure legend.

Line 428: Please correct “therefore”.

The correction has been made.

Authors contributions

Authorship is the collective responsibility of the authors, although it is my responsibility to allow you to appreciate the recommendations of the International Committee of Medical Journal Editors (ICMJE) on authorship criteria. Authors will meet four cumulative criteria, but none about “resources” contributions or “funding acquisition”. Collaborations are encouraged in every aspect of today’s life, but money should never be the reason for any collaboration.

All author meet the four cumulative criteria of the International Committee of Medical Journal Editors (ICMJE), including

  • Substantial contributions to the conception or design of the work; or the acquisition, analysis, or interpretation of data for the work; AND
  • Drafting the work or revising it critically for important intellectual content; AND
  • Final approval of the version to be published; AND
  • Agreement to be accountable for all aspects of the work in ensuring that questions related to the accuracy or integrity of any part of the work are appropriately investigated and resolved.

The critera “resources” and “funding acquisition” were included in accordance to Contributor Roles Taxonomy (CRediT) as required by MDPI. Next to “resources” and “writing - review and editing”, RS and GW also contributed to the formal analysis of the present work, which is now noted in the Author Contributions.

Please define the contributions “investigation” and visualization”.

We defined the contributions “investigation” and “visualization” according to Contributor Roles Taxonomy (CRediT) as required by MDPI:

Investigation: Conducting a research and investigation process, specifically performing the experiments, or data/evidence collection.

Visualization: Preparation, creation and/or presentation of the published work, specifically visualization/data presentation.

Please place funding in the funding section.

We now placed the funding in the funding section.

References section

Please, abbreviate the journal’s name of references 4; 6; 9; 10; 12; 13; 14; 17; 18; 22; 23; 30; 31, 37; and 40.

We corrected the journal’s names accordingly.

Reference 8: Please lowercase the K and the E from “BoneKEy”.

The correction has been made.
